# Cardiac Magnetic Resonance Imaging Based Ischemic Injury Pattern in Patients with Acute Myocardial Infarction Sensu Left Ventricular Global Systolic Function

**DOI:** 10.3390/diagnostics14060588

**Published:** 2024-03-11

**Authors:** Lyudmila Salyamova, Valentin Oleynikov, Natalia Donetskaya, Alexander Vdovkin, Angelina Chernova, Irina Avdeeva

**Affiliations:** 1Therapy Department, Penza State University, 440026 Penza, Russiaangelinakorneeva170498@gmail.com (A.C.);; 2Regional Clinical Hospital n/a N.N. Burdenko, 440026 Penza, Russia; enigmee@rambler.ru (N.D.); wdovkin@yandex.ru (A.V.)

**Keywords:** ST-segment elevation myocardial infarction, cardiac MRI, ischemic injury pattern, left ventricular global systolic function

## Abstract

The purpose of the study was to identify factors characterizing a decrease in LV global systolic function in patients with ST-segment elevation myocardial infarction (STEMI) after revascularization using cardiac magnetic resonance imaging (MRI)-based ischemic injury pattern and laboratory parameters sensu left ventricular global systolic function. A total of 109 STEMI patients were examined. The patients underwent contrast-enhanced cardiac MRI with a 1.5 Tesla GE SIGNA Voyager (GE HealthCare, Chicago, IL, USA) on the 7th–10th days from the onset of the disease. According to cardiac MRI analysis, the patients were divided into the following groups with regard to left ventricular ejection fraction (LVEF) values: Group 1—patients with LVEF ≥ 50%; group 2—patients with mildly reduced LVEF 40–49%; group 3—patients with low LVEF < 40%. A predominance of most parameters of the ischemic injury pattern was noted in patients with mildly reduced and low LVEF versus patient group with LVEF ≥ 50%. Some risk factors for a decrease in LVEF < 50% systolic function in STEMI patients after revascularization were revealed: male gender; time from the onset of the anginal attack to revascularization; coronary artery status; several LV parameters; ischemic injury characteristics; natriuretic peptide and troponin I levels.

## 1. Introduction

Chronic heart failure (CHF) is among the major complications of acute myocardial infarction (MI) and a typical outcome of the cardiovascular disease continuum, affecting disability and mortality worldwide. As reported by the Heart Failure Society of America, the incidence of post-MI CHF has not changed over the years, despite significant advances in treating this disease in the acute phase [1]. According to the Cardiovascular Disease in Norway (CVDNOR) Project data, 18.7% of patients experienced CHF during hospitalization for acute MI, 13% of patients—within 30 days of the index event, and 32.6% of patients—within a year of hospital discharge [2].

Left ventricular ejection fraction (LVEF) is the central measure of left ventricular contractile function, which determines the prognosis of patients in the post-MI period [3]. The frequency of systolic dysfunction ranges from 13–36% in the acute and subsequent MI stages, depending on its assessment criteria [4,5]. Wherein, being a significant predictor of mortality, reduced LVEF (<40%) is closely associated with an increase in the number of hospitalizations and low quality of life, as compared to preserved LVEF [6].

Currently, transthoracic echocardiography is used both for the initial assessment of ischemic injury and LV systolic and diastolic dysfunction [7]. An alternative to echocardiography is contrast-enhanced cardiac magnetic resonance imaging (MRI). This method allows for estimating LV volumes and functions with high accuracy. Cardiac MRI provides a potential method to characterize scar and peri-infarct zones (PIZ), interstitial edema, microvascular obstruction (MVO), and intramyocardial hemorrhage (IMH) (Figure 1) [8]. Assessing the viability of myocardial segments is vital in predicting the recovery of myocardial contractile function, LVEF, and survival after MI. According to the American Heart Association, the major advantage of contrast-enhanced cardiac MRI is the ability to assess the presence of subendocardial MI and to delineate the transmural extent of MI [9].

The purpose of the study was to identify factors characterizing a decrease in LV global systolic function in patients with ST-segment elevation myocardial infarction (STEMI) after revascularization using cardiac MRI-based ischemic injury pattern and laboratory parameters sensu left ventricular global systolic function.

## 2. Material and Methods

A total of 109 STEMI patients were examined in the Emergency Cardiology Department of the Regional Clinical Hospital n. a. N.N. Burdenko (Penza, Russia) in 2020–2022 (protocol code CONTRAST-2, NCT04347434 at clinicaltrials.gov accessed on 1 February 2024). The study protocol and the case report form were approved by the Local Ethics Committee of Penza State University. Written informed consent was obtained from all patients.

The study included patients who met the following criteria: aged 30–70 years; acute STEMI confirmed by an electrocardiogram, a diagnostically significant increase in troponin I, and the presence infarct-related artery determined by coronary angiography.

The exclusion criteria were as follows: Repeated or recurrent MI; hemodynamically significant stenosis (>30%) of the left main coronary artery; pronouced concomitant diseases; NYHA classes III–IV of CHF. All patients underwent coronary angiography. Primary percutaneous coronary intervention (PCI) on the infarct-related artery (IRA) was performed in 68 patients, and 41 patients experienced a pharmaco-invasive strategy. The revascularization method was chosen depending on time elapsed since pain onset and transport delay between patient location and PCI center [10].

The STEMI patients received treatment in compliance with current clinical practice guidelines.

The patients underwent contrast-enhanced cardiac MRI with a 1.5 Tesla GE SIGNA Voyager (GE HealthCare, USA) on the 7th–10th days from the onset of the disease. Being a paramagnetic contrast agent, Clariscan™ (gadoteric acid) (GE Healthcare, Oslo, Norway) was used to enhance visualization in MRI procedures. A non-contrast scan protocol involved anatomical slices; LV long-axis cine images in 2- and 4-chamber projections; LV short-axis cine images; and T1-, T2-, and T2*-mapping. In accordance with post-contrast MRI studies, two-dimensional (2D) myocardial delayed enhancement (MDE) cardiac imaging was obtained along the short axis at the 7th minute and along the long axis in 2- and 4-chamber projections at the 10th–12th minute. Post-contrast T1 mapping was conducted in the two-dimensional modified Look-Locker imaging (2D MOLLI) sequence at the 15th min. Scanning was done using MDE of fast-imaging employing steady-state acquisition (MDE FIESTA) sequences along the short axis at the 20th minute. Image processing was performed using the cvi42^®^ software platform, version 5.11 (Circle Cardiovascular Imaging Inc., Calgary, AB, Canada). The following LV indicators were determined: End-diastolic volume index (EDVI); end-systolic volume index (ESVI); LVEF; left ventricular mass index (LVMI); local contractility index (LCI). The ischemic injury pattern was analyzed by post-ischemic MDE, identifying scar zone and PIZ heterogeneity, MVO, and IMH. A quantitative assessment of necrosis and PIZ masses and sizes relative to LV myocardial mass as well as MVO and IMH masses and sizes relative to the scar tissue mass was carried out. The lesion depth was studied using the global contrast index (GCI) [11].

The severity of coronary artery lesion was determined by quantitative coronary angiography in accordance with the SYNTAX score scale.

N-Terminal pro-B-type natriuretic peptide (NT-proBNP) and high-sensitivity C-reactive protein (hsCRP) were assessed in the blood on the 7th–9th days. High-sensitivity troponin I (hsTI) was recorded three times, the maximum values of the indicator being presented in the paper.

Statistical data analysis was carried out using the STATISTICA 13.0 software package (StatSoft, Tulsa, OK, USA). Quantitative data were presented as the mean and standard deviation (M ± SD) with a normal distribution or as the median and interquartile range (Me (Q 25%; Q 75%)) with a nonparametric distribution. The significance of differences between parametric data was assessed using the Student’s *t*-test, and the significance of differences between nonparametric data was evaluated by the Mann–Whitney U test. Qualitative data were compared using the chi-squared test (*χ*^2^). The logistic regression analysis was used to identify variables predicting a decrease in LVEF. Statistical significance was assumed at *p* < 0.05.

## 3. Results

According to cardiac MRI analysis, the patients were divided into the following groups with regard to LVEF values: Group 1—patients with LVEF ≥ 50% (*n* = 74); group 2—patients with mildly reduced LVEF 40–49% (*n* = 27); group 3—patients with low LVEF < 40% (*n* = 8). There was no difference between the compared groups in certain anthropometric and anamnestic data (Table 1). The patients in group 3 were older than those in group 1. Female trial participants were only observed in the patient group with LVEF ≥ 50%.

According to coronary angiography data, hemodynamically significant stenosis was detected in 50% of cases within one coronary artery; 32%—within two coronary arteries; 18%—within three or more coronary arteries (group 1); 30%, 37%, and 33% (group 2); 25% (*p*_1–2;1,2–3_ > 0.05), 25% (*p*_1–2;1,2–3_ > 0.05), and 50% of cases (*p*_1–3_ = 0.032), respectively (group 3). When analyzing the SYNTAX score scale, the lowest number of points was observed in patients with LVEF ≥ 50% versus other groups. A lesion of the anterior descending artery (ADA) as the IRA was detected significantly more often in the group with low LVEF as compared to the group with preserved LVEF. In turn, atherothrombosis of the right coronary artery (RCA) was observed in 37.8% of patients only in group 1. Wherein, there were no cases of RCA atherothrombosis in group 3.

A pharmaco-invasive strategy was used as a reperfusion therapy in patients with LVEF < 40%. However, pain-to-needle time was 2.5 times longer in this patient group versus group 1. Pain-to-balloon time in patients with LVEF 40–49% exceeded that in group 1 with LVEF ≥ 50%.

According to cardiac MRI analysis, the compared groups differed in the majority of the analyzed structural and functional LV characteristics (Table 2). With comparable EDVI, the patients with LVEF ≥ 50% had the lowest ESVI values; the intermediate values were recorded in the group with LVEF 40–49%, and the highest values were observed in the patients with LVEF < 40%. Similar differences were found for LCI. LVMI predominated in patients with low LVEF as compared to the group with preserved LVEF.

When analyzing the ischemic injury pattern, the worst infarct zone characteristics were revealed (Table 2, Figure 2). The values of the scar tissue mass were minimal in group 1; intermediate values—in group 2; and maximum values—in group 3. The size of the scar tissue of the total myocardial mass (%) prevailed in patients with LVEF 40–49% and LVEF < 40% versus the patient group with LVEF ≥ 50%. As for PIZ characteristics, the differences were only found between patients with preserved and mildly reduced LVEF. The infarct zone, including the scar tissue and PIZ, occupied an average of 25.8% of the total myocardial mass in the group with LVEF ≥ 50%; whereas it was 1.6 times larger with LVEF 40–49%, and 2.1 times larger in the group with LVEF < 40%.

In contrast to the group with preserved LVEF, the majority of patients with low LVEF experienced MVO (Figure 3). However, when analyzing MVO mass, its values were minimal in group 1; intermediate values—in group 2; and maximum values—in group 3. The frequency of IMH detection prevailed in patients with mildly reduced LVEF versus patients with LVEF ≥ 50%.

Notable results were obtained when analyzing the GCI. The lowest values were found in group 1; intermediate values—in group 2; and the highest values—in group 3.

In the early period of MI, hsTI amounted to 23,719.6 (5497.0; 49,239.2) pg/mL in patients with LVEF ≥ 50%; 21,217.4 (8663.0; 84,972.0) pg/mL—in patients with LVEF 40–49%; 133,112.5 ± 98,672.5 pg/mL (*p*_1,2–3_ < 0.05)—in patients with LVEF < 40%. Similar differences were also noted for BNP. Its indicator was equal to 154.0 (61.2; 373.9) ng/mL in group 1; 274.4 (196.4; 703.7) ng/mL—in group 2; 669.3 (276.1; 700.0) ng/mL (*p*_1–3_ = 0.035)—in group 3. The groups did not differ in hsCRP levels.

Risk factors for a decrease in LVEF < 50% in patients with MI were identified using univariate linear regression analysis (Table 3). The largest regression coefficient *β* (>0.500) was observed for the scar tissue mass, for the total mass of ischemic injury and the size thereof of the total myocardium mass (%), for GCI, and LCI. A less pronounced relationship was noted for PIZ, MVO, and IMH characteristics. Gender, the number of points on the SYNTAX score scale, type of IRA, and the time interval from the onset of anginal attack to revascularization were also associated with unfavorable values of LV global systolic function. It should be noted that the method of revascularization had no prognostic value in reducing LVEF (*p* = 0.690). The laboratory parameters for NT-proBNP and hsTI levels evidenced an autonomous position thereof in reducing LVEF <50%.

## 4. Discussion

The infarct size and the lesion depth are among the key factors affecting the development of complications and mortality among STEMI patients in the post-infarction period. Therefore, a large number of clinical studies are aimed at the timely identification of potential patients with an unfavorable prognosis in order to optimize drug therapy and surgical treatment [12].

LVEF, estimated from transthoracic echocardiography data, has been widely used in routine practices as an important predictor of survival in patients with acute coronary syndrome. An indicator level of less than 40% is associated with death, which develops in 15% of patients six months after the index event [13]. Cardiac MRI is considered the “gold” standard for assessing such LV characteristics as volumes, LVEF, and myocardial mass. It is recommended for the diagnosis of CHF in patients with suboptimal visualization based on transthoracic echocardiography. High MRI resolution provides for highly accurate and reproducible quantification of the above parameters [7]. In the present study, an increase in ESVI and LCI in patients with LVEF < 40% was a common manifestation of a decrease in LV systolic function, and myocardial hypertrophy did not increase its contractility.

Cardiac MRI is the most advantageous technique among the methods for visualizing post-MI changes due to the unique characteristics of myocardial tissue, high resolution, and the ability to quantify myocardial injury. In patients with MI, cardiac MRI can be used to determine the size of the infarction and the intact myocardium, MVO, and IMH, i.e., the main significant prognostic injury markers [8,11,14].

Diagnosis of infarction using MRI is based on gadolinium kinetics, which only penetrates through damaged membranes of cardiomyocytes and accumulates in the extracellular space. This provides an opportunity to detect and quantify various dysfunctional areas of the myocardium [9]. Thus, the gadolinium-based contrast agents wash out slowly from the myocardium as a result of the destruction of the cell membrane in acute MI. As for the scar stage, the gadolinium-based contrast agents wash out due to an increase in the interstitial space inside the scar [12].

The sizes of infarct zones visualized by MRI were calculated in grams or as a percentage of the LV mass. Local contrast enhancement can be used to detect infarcts as small as 1 g, subendocardial infarction being an example [15]. In the present study, the scar tissue mass in the patient groups with LVEF 40–49% and LVEF < 40% was 2.6 times and 4 times, respectively, larger than that in the patients with preserved LVEF.

Post-ischemic myocardial injury is morphologically heterogeneous. In addition to the infarct core, its structure includes PIZ, an area next to the intact myocardium, consisting of necrotic, ischemic, and intact cardiomyocytes [16]. In this study, the patients with mildly reduced LVEF were distinguished by higher values of PIZ mass and the size of the total LV myocardium mass versus group 1. According to Jensch P.J. et al., PIZ volume > 14 mL was considered a predictor of major adverse cardiovascular events such as death, recurrent MI, and congestive heart failure within one year after MI [17]. According to Jones R.E. et al., in patients with stable coronary artery disease (CAD), PIZ and infarct core masses were independent predictors of sudden cardiac death (SCD) [18]. As stated by Yan et al. [19], being a heterogeneous domain and having an arrhythmogenic potential, the peri-infarct zone is a powerful predictor of all-cause and cardiovascular disease mortality.

MVO, or the no-reflow phenomenon, is due to microvasculature vasoconstriction comorbid with distal embolization by atherosclerotic plaque elements, fibrin particles, platelets, and erythrocytes [20]. MVO is a predictor of recurrent cardiovascular events and a marker of subsequent adverse LV remodeling [20,21], and it develops in almost half of patients undergoing successful primary PCI [21].

MVO prevents adequate tissue perfusion despite the restoration of blood flow in the epicardial artery. This phenomenon is defined as hypoenhancement within the hyperintense infarct core (late gadolinium enhancement (LGE) regions) [22].

In MI patients with LVEF > 50% who underwent primary PCI, the presence and severity of MVO were associated with the development of fatal and non-fatal cardiovascular events during the 5-year monitoring period [23]. Notably, MVO volume ≥ 2.6% of LV mass improved long-term risk stratification of adverse outcomes after STEMI compared with such standard MRI indicators as LV volumes and LVEF [24].

IMH indicates more severe reperfusion injury in acute MI, resulting from extravasation of red blood cells through damaged vascular endothelium, damage to the coronary microvasculature, and disruption of vascular integrity [25,26]. IMH develops in 40% of revascularized patients with STEMI [27]. It is assumed that IMH leads to the accumulation of residual iron in the MI area, causing a cytotoxic effect on the myocardium and local inflammatory responses [25].

The combination of MVO and IMH is a predictor of decreased LVEF in patients with primary STEMI. In addition, the correlation analysis results revealed a relationship between the area of IMH based on MRI, and LV systolic dysfunction based on echocardiography [28]. When studying the relation between MRI findings and adverse outcomes (CHF and SCD), it was found that infarct size, MVO, IMH, and low LVEF are related to major adverse cardiac events and provide the prognostic value [29].

According to the results of this study, the frequency of MVO and its size relative to the scar tissue mass (%) prevailed in group 3 versus group 1. In addition, a progressive increase in MVO mass was revealed with a decrease in systolic LVEF. When comparing IMH, lesser variations were obtained. In particular, IMH was identified 2.4 times more frequently in patients with LVEF 40–49% versus patients with LVEF ≥ 50%. There were no differences recorded in the quantitative characteristics of this parameter.

The GCI serves as an indicator of the injury depth in MRI [8]. As the GCI increases, there is a decrease in the likelihood of improving the contractility of myocardial segments. The infarct-containing segments with > 50% transmural extent of infarction have a minor opportunity to restore segmental contractile function, despite successful coronary revascularization [9]. An increase in the frequency of MVO and higher values of myocardial necrosis markers were noted in the patient group with a high GCI level [11]. Demirkiran et al. [30] observed a significant relationship between GCI, time to reperfusion, and LVEF. The pain-reperfusion time delay leads to progressive cardiomyocyte death from the endocardium to the epicardium and may indicate the wavefront phenomenon of ischemic cell death [30]. In this study, the severity of systolic dysfunction was associated with progressive deterioration of GCI and an increase in pain-to-needle time and pain-to-balloon time. The level of LVEF < 40% was characterized by the most adverse parameter values.

Key diagnostic biomarkers used in routine clinical practice for patients with various CAD forms include cardiospecific isoforms of troponins and natriuretic peptides. Troponin I is recommended to be determined both for the diagnosis of myocardial injury in acute coronary syndrome and for assessing the risk of adverse cardiovascular events in the general population [31]. A number of studies have revealed a positive correlation between the mass of scar tissue, PIZ, and MVO according to MRI and the level of troponin I in patients with MI [11,32]. According to MRI, the concentration of both troponin I and NT-proBNP strongly correlates with the size of myocardial infarction [33]. A close relationship between NT-proBNP in blood plasma, MVO, and GCI was highlighted by Bruder et al. [34]. In patients without cardiovascular disease at the time of their inclusion in the study, the initially higher NT-proBNP was associated with a decrease in LVEF and the risk of myocardial scar formation according to MRI after 10 years of observation [35]. We have established the predominance of both hsTI and NT-proBNP in the group of patients with LVEF < 40%.

## 5. Conclusions

In this study, the presence of LV systolic dysfunction in patients with primary STEMI was distinguished by a natural deterioration of structural and volumetric parameters and a decrease in the efficiency of cardiac contraction according to MRI.

A predominance of most parameters of the ischemic injury pattern was noted in patients with mildly reduced and low LVEF versus the patient group with LVEF ≥ 50%. Differences in the ischemic injury mass were recorded between the groups with LVEF 40–49% and LVEF < 40% due to scar tissue, MVO mass, LCI, and GCI, which indicates transmural injury.

The following risk factors for a decrease in LVEF < 50% systolic function in STEMI patients after revascularization were revealed: male gender; time from the onset of the anginal attack to revascularization; coronary artery status; ESVI; LVMI; ischemic injury characteristics, including scar tissue and PIZ; presence and severity of MVO; IMH mass; global contrast and local contractility indices; NT-proBNP and hsTI levels.

## Figures and Tables

**Figure 1 diagnostics-14-00588-f001:**
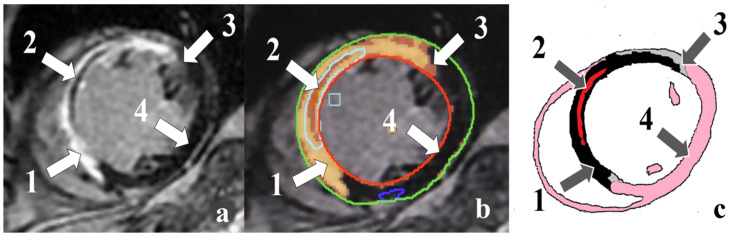
Cardiac MRI-based ischemic injury pattern (own observation): (**a**)—myocardial delayed enhancement; (**b**)—post-processing; (**c**)—schematic representation. Infarct zones: 1—necrosis; 2—microvascular obstruction; 3—peri-infarct zone heterogeneity; 4—intact myocardium.

**Figure 2 diagnostics-14-00588-f002:**
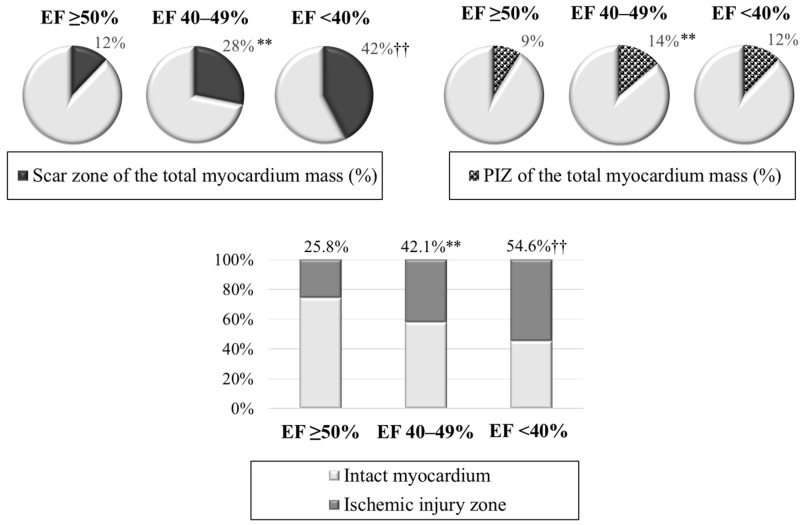
Characteristics of ischemic injury zone in the compared groups. Note: ** *p* < 0.01—significant differences between group 1 and group 2; †† *p* < 0.01—significant differences between group 1 and group 3; LVEF—left ventricular ejection fraction; PIZ—peri-infarct zone heterogeneity.

**Figure 3 diagnostics-14-00588-f003:**
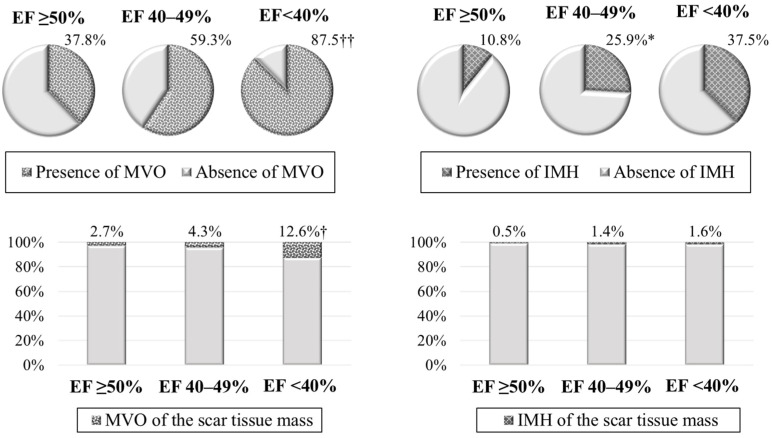
Characteristics of reperfusion injury in the compared groups. Note: * *p* < 0.05—significant differences between group 1 and group 2; † *p* < 0.05, †† *p* < 0.01—significant differences between group 1 and group 3; IMH—intramyocardial hemorrhage; LVEF—left ventricular ejection fraction; MVO—microvascular obstruction.

**Table 1 diagnostics-14-00588-t001:** Comparative characteristics of groups.

Parameter	LVEF ≥ 50% (*n* = 74)	LVEF 40–49% (*n* = 27)	LVEF < 40% (*n* = 8)	*p*
	Group 1	Group 2	Group 3	
Age, years	57 (50; 61)	58 (54; 62)	61.8 ± 6.0	*p*_1–3_ = 0.013
Male/female, *n* (%)	62/12 (84/16)	27/0 (100/0)	8/0 (100/0)	*p*_1–2_ = 0.026
Body mass index, kg/m^2^	27.9 ± 3.9	27.5 ± 3.9	26.5 (23.3; 29.2)	ns
Systolic blood pressure, mm Hg	130 (122; 135)	131.5 ± 15.4	129.3 ± 24.8	ns
Diastolic blood pressure, mm Hg	80 (78; 90)	79.8 ± 9.4	82.7 ± 14.7	ns
Anamnesis of CAD, *n* (%)	19 (25.7)	8 (29.6)	2 (25.0)	ns
Arterial hypertension, *n* (%)	62 (83.8)	24 (88.9)	6 (75.0)	ns
SYNTAX score, points	15.3 ± 7.1	19.7 ± 10.3	21.0 ± 9.9	*p*_1–2_ = 0.017 *p*_1–3_ = 0.042
Infarct-related artery:				
ADA, *n* (%)	32 (43.2)	15 (55.6)	7 (87.5)	*p*_1–3_ = 0.017
RCA, *n* (%)	28 (37.8)	7 (25.9)	0 (0)	*p*_1–3_ = 0.032
Other artery, *n* (%)	14 (19.0)	5 (18.5)	1 (12.5)	ns
Pharmaco-invasive revascularization/primary PCI, *n* (%)	27/47 (36.5/63.5)	8/19 (29.6/70.4)	6/2(75.0/25.0)	*p*_1–3_ = 0.035 *p*_2–3_ = 0.021
Pain-to-needle time (pharmaco-invasive revascularization), min	90 (50; 120)	135.0 ± 79.3	200.0 (120; 270)	*p*_1–3_ = 0.004
Pain-to-balloon time (pharmaco-invasive revascularization/Primary PCI), min	223 (145; 330)	305 (210; 560)	305.0 (227.5; 432.5)	*p*_1–2_ = 0.019

Note: ADA—anterior descending artery; CAD—coronary artery disease; LVEF—left ventricular ejection fraction; RCA—right coronary artery; PCI—percutaneous coronary intervention. The data are presented as M ± SD with normal distribution and Me (Q 25%; Q 75%)—with nonparametric distribution; *p*—statistical significance; ns—non-significant differences.

**Table 2 diagnostics-14-00588-t002:** Cardiac MRI indicators in the compared groups.

Parameter	LVEF ≥ 50% (*n* = 74)	LVEF 40–49% (*n* = 27)	LVEF < 40% (*n* = 8)	*p* _1–2_	*p* _1–3_	*p* _2–3_
	Group 1	Group 2	Group 3			
Standard cardiac MRI indicators						
EDVI, mL/m^2^	76.8 (67.0; 86.0)	79.6 ± 14.8	90.7 ± 24.0	0.620	0.169	0.069
ESVI, mL/m^2^	33.4 (28.3; 38.3)	41.6 ± 10.1	58.8 ± 18.8	<0.001	<0.001	0.002
LVMI, g/m^2^	57.9 (51.9; 69.5)	60.1 (56.1; 71.8)	73.9 ± 17.6	0.0936	0.018	0.121
LVEF, %	55.5 (52.6; 59.2)	47.0 (44.1; 49.0)	37.4 (34.7; 38.3)	<0.001	<0.001	<0.001
LCI	1.4 (1.1; 1.6)	1.9 ± 0.4	2.6 ± 0.4	<0.001	<0.001	<0.001
Cardiac MRI-based parameters of ischemic injury						
Scar tissue mass, g	15.7 (5.6; 24.7)	41.1 (13.6; 56.4)	62.5 ± 35.9	<0.001	<0.001	0.036
PIZ mass, g	10.4 (5.8; 16.5)	17.6 ± 7.7	17.0 ± 8.0	<0.001	0.089	0.839
Ischemic injury mass, g	27.8 (10.8; 39.8)	53.9 ± 24.4	79.5 ± 35.1	<0.001	<0.001	0.025
MVO mass, g	0.0 (0.0; 1.0)	1.0 (0.0; 3.1)	7.3 ± 6.8	0.028	<0.001	0.011
IMH mass, g	0.0 (0.0; 0,1)	0.0 (0.0; 0.5)	0.0 (0.0; 3.5)	0.221	0.167	0.421
GCI, %	17.7 (11.8; 29.4)	34.2 ± 12.2	64.0 (41.2; 66.9)	<0.001	<0.001	0.007

Note: EDVI—end-diastolic volume index; ESVI—end-systolic volume index; GCI—global contrast index; IMH—intramyocardial hemorrhage; LCI—local contractility index; LVEF—left ventricular ejection fraction; LVMI—left ventricular mass index; MRI—magnetic resonance imaging; MVO—microvascular obstruction; PIZ—peri-infarct zone heterogeneity. The data are presented as M ± SD with normal distribution and Me (Q 25%; Q 75%)—with nonparametric distribution; *p*—statistical significance.

**Table 3 diagnostics-14-00588-t003:** Risk factors for a decrease in LVEF < 50% in patients with acute MI based on univariate linear regression analysis data.

Parameter	β	b	*p*
Gender	0.242	0.361	0.011
Pain-to-needle time, min	0.482	0.003	0.002
Pain-to-balloon time, min	0.195	0.001	0.042
SYNTAX score, points	0.262	0.015	0.006
Lesion of ADA, RCA, or any other artery as infarct-related one	0.197	0.104	0.039
ESVI, mL/m^2^	0.417	0.016	<0.001
LVMI, g/m^2^	0.230	0.008	0.016
LCI	0.574	0.017	<0.001
Scar tissue mass, g	0.549	0.012	<0.001
Scar zone of the total myocardium mass, %	0.547	0.017	<0.001
PIZ mass, g	0.329	0.019	<0.001
PIZ of the total myocardium mass, %	0.282	0.022	0.003
Ischemic injury mass, g	0.548	0.009	<0.001
Ischemic injury of the total myocardium mass, %	0.529	0.013	<0.001
Presence of MVO	0.261	0.244	0.006
MVO mass, g	0.382	0.065	<0.001
MVO of the scar tissue mass, %	0.259	0.019	0.006
IMH mass, g	0.266	0.120	0.005
Global contrast index, %	0.664	0.628	<0.001
HsTI, pg/mL	0.237	0.001	0.006
NT-proBNP, pg/mL	0.325	0.001	0.013

Note: ADA—anterior descending artery; ESVI—end-systolic volume index; HsTI—high-sensitivity troponin I; IMH—intramyocardial hemorrhage; LCI—local contractility index; LVEF—left ventricular ejection fraction; LVMI—left ventricular mass index; MI—myocardial infarction; MVO—microvascular obstruction; NT-proBNP—N-terminal pro-B-type natriuretic peptide; PCI—percutaneous coronary intervention; PIZ—peri-infarct zone heterogeneity; RCA—right coronary artery; β—regression coefficient; b—angular coefficient; *p*—statistical significance.

## Data Availability

The data presented in this study are available on request from the corresponding author. The data are not publicly available due to ethical reasons.

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
