# Peer review of "Cardiac Magnetic Resonance Imaging Based Ischemic Injury Pattern in Patients with Acute Myocardial Infarction Sensu Left Ventricular Global Systolic Function"

_diagnostics, 2024, doi:10.3390/diagnostics14060588_

Round 1

Reviewer 1 Report

Comments and Suggestions for Authors

Dear Authors,

Firstly, I understand that the work is interesting and will certainly contribute to the area in question. In order to make the article even better, I make the following suggestions:

1) Check that all acronyms are properly preceded with their respective meaning in full;

2) Correct some data in table 2 that is unformatted;

3) Review the text of the article so that small typing errors can be corrected.

Kind regards,

Comments on the Quality of English Language

Minor editing of English language required.

Author Response

Response to Reviewer 1 Comments

Point-by-point response to Comments and Suggestions for Authors

  • Check that all acronyms are properly preceded with their respective meaning in full;

Corrected (page 1, introduction, paragraph 2, line 1)

  • Correct some data in table 2 that is unformatted;

Corrected (page 4)

  • Review the text of the article so that small typing errors can be corrected.

Corrected

Kind regards,

Reviewer 2 Report

Comments and Suggestions for Authors

Interesting study.  I have several questions:

How was it determined who received PCI and who received pharmacological treatment

How did type of treatment impact results

Was there any explanation for the difference in resluts between sexes

Why was there such a difference in time to treatment intervention between the PCI and pharmacological and dis this effect your results

Comments on the Quality of English Language

The English is very aceptable

Author Response

Response to Reviewer 2 Comments

Point-by-point response to Comments and Suggestions for Authors

  • How was it determined who received PCI and who received pharmacological treatment

 We were guided by the recommendations of the European Society of Cardiology for the management of patients with acute myocardial infarction with ST segment elevation (2017). According to them, if the delay time revascularization did not exceed 120 minutes, the patient underwent primary percutaneous coronary intervention. If it was not possible to perform primary percutaneous coronary intervention in the target terms, then pre-hospital thrombolytic therapy was carried out followed by hospitalization in invasive center for percutaneous coronary intervention.

Added: The revascularization method was chosen depending on time elapsed since pain onset and transport delay between patient location and PCI centre [10]. (page 2, Material and Methods, paragraph 2, lines 5-7)

  • How did type of treatment impact results

For this sample size, there is no association between treatment type and left ventricular ejection fraction installed.

Added: It should be noted that the method of revascularization had no prognostic value in reducing LVEF (p=0.690). (page 6, paragraph 3, lines 8-9).

  • Was there any explanation for the difference in resluts between sexes

 Our study included a limited number of women. That is, identified the differences are due to the relatively small sample.

  • Why was there such a difference in time to treatment intervention between the PCI and pharmacological and dis this effect your results

Pain-to-needle time included only patients with a pharmaco-invasive revascularization strategy. Pain-to-balloon time is indicated for patients with both primary PCI and pharmaco-invasive strategy. The corresponding explanation is included in the table (Table 1).

Kind regards,